# 7 Days Later: Analyzing Phishing-Site Lifespan After Detected

## Abstract

Phishing attacks continue to be a major threat to internet users, causing data breaches, financial losses, and identity theft. This study provides an in-depth analysis of the lifespan and evolution of phishing websites, focusing on their survival strategies and evasion techniques. We analyze 286,237 unique phishing URLs over five months using a custom web crawler based on Puppeteer and Chromium. Our crawler runs on a 30-minute cycle, systematically checking the operational status of phishing websites by collecting their HTTP status codes, screenshots, HTML, and HTTP data. Temporal and survival analyses, along with statistical tests, are used to examine phishing website lifecycles, evolution, and evasion tactics. Our findings show that the average lifespan of phishing websites is 54 hours (2.25 days) with a median of 5.46 hours, indicating rapid takedown of many sites while a subset remains active longer. Interestingly, logistic-themed phishing websites (*e.g.*, USPS) operate within a compressed timeframe (1.76 hours) compared to other brands (*e.g.*, Facebook). We further analyze detection effectiveness using Google Safe Browsing (GSB). We find that GSB detects only 18.4% of phishing websites, taking an average of 4.5 days. Notably, 83.93% of phishing sites are already taken down before GSB detection, meaning GSB requires more prompt detection. Moreover, 16.07% of phishing sites persist beyond this point, surviving for an additional 7.2 days on average, resulting in an average total lifespan of approximately 12 days. We reveal that DNS resolution error is the main cause (67%) of phishing website takedowns. Finally, we uncover that phishing sites with extensive visual changes (more than 100 times) exhibit a median lifespan of 17 days, compared to 1.93 hours for those with minimal modifications. These results highlight the dynamic nature of phishing attacks, the challenges in detection and prevention, and the need for more rapid and comprehensive countermeasures against evolving phishing tactics.

## 1 Introduction

Phishing remains one of the most pervasive threats in the web ecosystem, causing data breaches, financial losses, identity theft, operational disruptions, and harm to reputation [1, 2]. A recent FBI report [3] reveals that phishing-related financial damages surpassed $10 billion in 2022—an increase of $4 billion from the previous year.

Phishing attacks use deceptive websites that closely mimic legitimate platforms like financial institutions and social media sites (*e.g.*, PayPal and Facebook) to trick users into disclosing sensitive information, such as login credentials and financial data. The lifecycle of a phishing attack ranges from the launch of these fraudulent sites to their detection (*e.g.*, blocklisted by Google Safe Browsing (GSB) [4]) or takedown by security authorities.

It seems that phishing websites may have shorter, more variable lifecycles than benign sites, ranging from hours to several weeks. Understanding these patterns and their causes is key to developing better defense strategies. Insights can help identify intervention points, anticipate new tactics, design targeted user awareness programs, and optimize detection and response resources.

Prior research has made attempts to analyze the lifecycle of phishing attacks [5–8]. Notably, Oest *et al.* [5] observed that the average lifespan of a phishing campaign spans approximately *21 hours* from the first to the last victim visit. However, their dataset for the study only contains PayPal-related phishing attacks (*i.e.*, phishing websites load resources (*e.g.*, logo images) from PayPal's servers, where the websites' domains are *not known to PayPal*), meaning that their findings might be biased for certain types of phishing attacks. In particular, Lim *et al.* [9] discover that a majority of phishing sites did *not* rely on targeting-brand hosted resources. It motivates us to challenge the assumption that most phishing attacks leverage targeting-brand hosted resources, suggesting the need for a more comprehensive analysis of phishing lifespans. Furthermore, McGrath [6] and Drury [7] analyzed temporal patterns of phishing URLs and domains, examining registration timelines, URL characteristics, and hosting infrastructure. However, their study did not extensively analyze the content or structure of phishing pages, find a cause of phishing websites' takedown, or identify characteristics of the phishing lifecycle after detection. It also motivates us to focus on the *actual content and structure* of phishing websites.

Our research aims to address these gaps by conducting a comprehensive, data-driven analysis of the phishing ecosystem (particularly the lifespan of phishing attacks). We utilize a unique dataset collected every 30 minutes from phishing websites identified by the Anti-Phishing Working Group (APWG) [10], allowing us to examine a diverse range of phishing websites. Specifically, our study employs a multifaceted methodology, integrating temporal analysis, web traffic data, and screenshots to scrutinize phishing lifespans. By examining diverse features such as DNS records, HTML content, and visual elements, we offer a comprehensive understanding of modern phishing lifecycles, infrastructure, and evasion tactics. This approach allows us to delve deep into the mechanics of phishing operations, addressing critical research questions that previous studies have not fully explored.

We first examine the lifespan of phishing websites (**RQ1: How long do phishing websites remain active, and how does this vary across different targeted brands?**). This investigation uncovers intriguing patterns in phishing operations, takedown efforts, and the effectiveness of GSB detection, challenging conventional assumptions about phishing lifespans and proposing new strategies for improved detection and mitigation. Diving deeper into the technical underpinnings of phishing websites, we explore the infrastructure changes throughout phishing websites' lifetime (**RQ2: What factors cause phishing websites to be taken down?**). By analyzing DNS records, IP addresses, and hosting patterns, we uncover sophisticated tactics employed by phishing attackers to evade detection and extend the lifespan of their websites. Finally, we explore the dynamic evolution of phishing websites (**RQ3: What technical measures do phishing websites employ to avoid detection, and how do these change over a website's lifetime?**), including an examination of DOM structure, third-party script usage, and various anti-detection techniques.

By addressing these research questions, our study aims to provide insights into the dynamics of phishing ecosystems. Specifically, this paper aims to investigate a recent advancement of phishing threats via large-scale fine-grained data collection to reveal the complex lifecycle of phishing websites. Our contributions are as follows:

- We present a comprehensive analysis of phishing website life-cycles using a unique, high-frequency dataset collected at 30-minute intervals over five months. This reveals critical insights into phishing website durations and detection timelines. (1) The average lifespan of a phish is 2.25 days, indicating a significant gap in real-time protection against phishing threats over three days. (2) We observe that logistics-themed phishing attacks (*e.g.*, USPS or DHL) tend to operate within shorter timeframes compared to those targeting other sections (*e.g.*, Facebook).
- We reveal that GSB takes an average of 4.5 days to detect phishing sites, with only 18.41% of sites in our dataset being detected. Moreover, we observe a vulnerability window where 83.93% of phishing sites are taken down before GSB detection.
- We find that phishing sites with extensive visual changes (more than 100 times) have a median lifespan of 17 days, compared to just 1.93 hours for those with minimal modifications, demonstrating the effectiveness of visual alterations in prolonging phishing operations.
- We have open-sourced our data collection framework to promote transparency and reproducibility. The code will be made publicly available upon acceptance.

## 2 Background

**Phishing Attacks.** Phishing attacks are social engineering tactics employed by cybercriminals to trick individuals into disclosing sensitive information, such as login credentials or credit card details. The attacks typically involve the creation of fraudulent websites designed to closely resemble legitimate platforms, such as online banking services or social media sites (*e.g.*, Facebook). The phishing attack lifecycle begins with the launch of a deceptive website and persists until the site is either taken down by attackers or removed by security authorities. During this attack period, attackers can collect sensitive information (*e.g.*, credentials) from victims.

**Detecting and Mitigating Phishing Attacks.** The current anti-phishing ecosystem (*e.g.*, GSB) relies heavily on blocklist-based approaches, which play a crucial role in mitigating the impact of phishing sites by facilitating their quick identification and blocking [11]. Blocklist systems function by first collecting potential phishing URLs, then verifying their legitimacy through analysis, and finally adding confirmed phishing URLs to the blocklist to actively block user access and protect against malicious sites. GSB [12] is a widely adopted blocklist, integrated into major browsers like Google Chrome, Apple Safari, and Firefox.

**Evasion Techniques.** Phishing attackers develop sophisticated evasion techniques to prolong the lifespans of their phishing attacks [13]. This deception is implemented through client- or server-side code, utilizing filters based on various attributes. Attackers also adopt URL manipulation strategies to evade detection. Benign URLs redirect victims to landing pages containing deceptive keywords [14]. This technique undermines URL-based heuristic detection methods [15] and complicates the process of correlating URLs

within the same redirection chain [16]. Additionally, phishing attackers can set DNS TTLs to facilitate fast-flux service networks [17] as low as zero seconds, effectively disabling caching [18].

## 3 Motivation

Prior research [5–8], despite their valuable insights and observations, have limited dataset scope. In particular, [5] focused on victim traffic to phishing websites targeting a single organization. Moreover, existing anti-phishing strategies often rely on static, point-in-time analyses, leaving critical gaps in our understanding of how these threats adapt and persist over time. We recognize the need for a more dynamic, ecosystem-wide approach to studying phishing campaigns. The importance of analyzing phishing site lifespans becomes evident when considering recent attack patterns. For instance, in a sophisticated phishing campaign targeting Instagram, attackers evaded detection by changing redirected URLs nine times within a 24-hour period (see Section 5.1). This rapid evolution starkly underscores the challenges faced by existing blocklist systems. For blocklisting to be effective, it must detect and respond to these sites before attackers can switch domains or URLs. This task becomes increasingly difficult with such agile evasion tactics.

Our study provides crucial insights into the typical detection times needed for blocklist systems to respond effectively against rapidly evolving phishing threats. By employing a high-frequency data collection methodology, we capture the dynamic nature of phishing websites, offering unprecedented insights into their lifecycles, adaptation strategies, and resilience mechanisms. This approach allows us to identify critical timeframes for anti-phishing measures to operate effectively. By analyzing these timeframes, we can develop practical solutions to reduce detection and response times within critical windows significantly. This knowledge is essential for blocking phishing sites before they can fully execute their attack strategies and developing proactive, adaptive defense mechanisms to keep pace with evolving threats.

## 4 Our Crawler Design for Data Collection

**Phishing URL Source.** To address our research questions, we leverage the APWG eCX platform [10], one of the widely used repositories in previous research [5, 9, 13, 19–24]. APWG eCX aggregates phishing URLs reported from a wide range of sources, including security vendors, financial institutions, and Internet Service Providers (ISPs). It provides real-time updates on active and reliable phishing websites. Note that since APWG eCX provides only metadata (*e.g.*, phishing URLs and target brands), we develop a custom crawler that periodically monitors phishing websites and assesses their operational status (*e.g.*, take down or still alive).

**Web Crawler Design.** Figure 1 illustrates our data collection system. At its core, it uses a Redis-based queueing system to manage input URLs. Then, we implement a custom web crawler using Puppeteer [25] and Chromium, augmented with stealth plugins [26] to bypass potential anti-bot measures employed in sophisticated phishing websites. The system is capable of processing approximately 250 URLs per minute with 16 parallel browsing instances. It runs on a 30-minute cycle, systematically checking the operational status of phishing websites by collecting their HTTP status codes (*e.g.*,

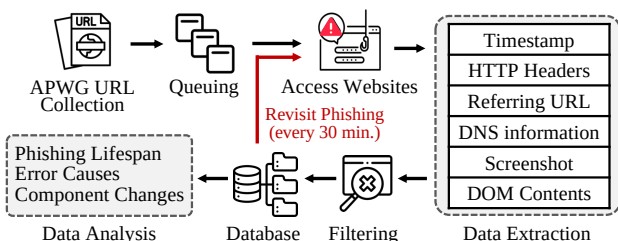

**Figure 1: Overview of Data Collection.**

HTTP 404 error). If a website returns an error code *three consecutive times* or is confirmed offline, we stop visiting it in the next cycle.

Our crawler collects the following comprehensive data from each visited site to facilitate in-depth analysis and tracking of phishing infrastructure and techniques: (1) network information such as IP addresses and WHOIS data, (2) full-page screenshots (with a 1280x960 viewport) and their color histogram, (3) the complete HTML contents (including all dynamically generated elements), (4) the HTTP data (including request headers, content policy, referrer information, and HTTP version), and (5) all redirected links (the entire URL chain from the original APWG link to the landing page). **Error Handling and Resilience.** To ensure data integrity and completeness, we carefully handle various errors. In case of failure, each URL is given up to three attempts, and failures are categorized using a comprehensive error classification system. Specifically, as each attempt is made every 30 minutes, if three attempts fail in a row (*i.e.*, unable to connect for 90 minutes), the URL is excluded from the data for the next visits.

**URLs Blocklisted by Google Safe Browsing (GSB).** GSB [12] is one of the most popular blocklist-based anti-phishing systems, integrated into many modern web browsers such as Google Chrome, Firefox, and Apple Safari [27–29].[1] We leverage GSB Update APIs v4 [31] to collect GSB's blocked phishing URLs. GSB stores the URLs into SHA-256 hashes rather than in plain text [31]. We collect 10,249,563 URLs from GSB during the same collection period of phishing URLs.

**Our Collected Dataset Overview.** Our crawler processes 286,237 unique phishing URLs from the APWG feed system and a total of 2,742,542 (2.7M+) visits. This indicates an average of 9.58 visits per URL ($\sigma = 45.2$), reflecting our multiple crawling attempts and the dynamic nature of phishing sites. The substantial number of feed URLs and the high volume of collected traffic underscores the pervasive nature of phishing attempts during our study period. The high average of URL traffic per site suggests that many phishing websites remain active for extended periods (287.4 minutes or about 4.79 hours), with 38,911 sites (25% of the total) receiving more than 18 visits, indicating prolonged activity.

## 5 Lifespan of Phishing Sites (RQ1)

To answer our first research question "**How long do phishing websites remain alive even after detected, and how does this vary across different targeted brands?**", we conduct a series of analysis: (1) calculating the overall lifespan of phishing sites across different brands, (2) determining the detection times by GSB, (3) comparing the lifespans and detection times across various

[1]According to Google's report, five billion users benefit from GSB's warning [30].

**Table 1: Lifespan of Top 10 Phishing Brands.**

| Brands | # URL (%) | Avg. Lifespan | Med. | Max. | Std. |
|---|---|---|---|---|---|
| Facebook | 77,525 (27.08) | 56.57 h. (2.36 d.) | 8.68 h. | 33.31 d. | 4.71 d. |
| USPS | 32,089 (11.21) | 44.08 h. (1.83 d.) | 1.76 h. | 31.15 d. | 4.43 d. |
| AT&T | 9,811 (3.42) | 41.15 h. (1.71 d.) | 9.37 h. | 31.30 d. | 3.70 d. |
| WhatsApp | 7,261 (2.53) | 41.47 h. (1.73 d.) | 5.80 h. | 31.38 d. | 4.18 d. |
| Instagram | 4,746 (1.66) | 54.12 h. (2.55 d.) | 12.62 h. | 31.01 d. | 4.43 d. |
| DHL | 3,633 (1.27) | 61.95 h. (2.58 d.) | 1.89 h. | 31.26 d. | 5.82 d. |
| SwissPass | 1,912 (0.67) | 66.57 h. (2.77 d.) | 5.89 h. | 30.05 d. | 5.17 d. |
| Evri | 1,104 (0.39) | 73.85 h. (3.08 d.) | 1.71 h. | 30.22 d. | 6.33 d. |
| Rakuten | 604 (0.21) | 66.29 h. (2.76 d.) | 4.73 h. | 29.61 d. | 4.67 d. |
| Google | 582 (0.20) | 49.26 h. (2.05 d.) | 6.11 h. | 26.95 d. | 3.97 d. |
| **Total** | 286,237 (100) | 54.04 h. (2.25 d.) | 5.46 h. | 38.43 d. | 4.81 d. |

∗ **Facebook** includes Meta; **Total** are also included 1,654 brands.
† **h.** indicates 'hours'; **d.** indicates 'days.'

targeted brands, and (4) identifying patterns and factors influencing phishing site longevity. From this analysis, we aim to obtain insights regarding the persistence of phishing sites and the effectiveness of detection mechanisms across different brand targets.

### 5.1 Overall Phishing Lifespan

**Top Target Brands.** In our dataset, Facebook (including Meta) emerges as the most frequently targeted brand for phishing attacks, representing 27.08% of the total URLs (77,525 out of 286,237), as shown in Table 1. We also compare the top brands used in phishing attacks measured in previous studies [9, 19, 21, 23] that utilized an older dataset from the APWG eCX repository. Interestingly, we find that USPS, which is ranked in the second position in Table 1, was never ranked within the top 10 in previous studies, indicating a shift in the phishing landscape. This change highlights the dynamic nature of phishing trends and the importance of continuous monitoring and analysis.

**Phishing Website Lifespan.** As shown in Table 1, the overall average lifespan of phishing websites in our dataset is 54.04 hours (2.25 days), with a median of 5.46 hours. This significant difference between the average and median lifespans reveals a highly skewed distribution, as shown in Figure 2. Specifically, 50% of phishing sites are swiftly taken down within 5.46 hours. However, a small subset of sites remain active for longer periods, leading to a longer average lifespan. For example, a Microsoft-themed phishing site lasts its operation for 38.43 days. The persistence of some sites could be attributed to various factors: they might be hosted in jurisdictions where legal action is challenging, employ sophisticated evasion techniques, or be overlooked due to their low profile or targeting of smaller, less-resourced organizations.

**Lifespan by Top Target Brand.** Further analysis, categorized by targeted brand, reveals that phishing websites' lifespan varies across impersonated companies. The average lifespan ranges from 41.15 hours (AT&T) to 73.85 hours (Evri). Interestingly, logistic companies such as DHL, USPS, and Evri show distinct patterns in the lifespans of their phishing sites. These brands tend to have shorter median lifespans than other sectors (*e.g.*, Facebook): USPS with 1.76 hours, Evri with 1.71 hours, and DHL with 1.89 hours. In contrast, Facebook's lifespans are significantly longer, with a median of 8.68 hours. This pattern suggests that phishing attacks mimicking logistics companies operate within a compressed timeframe, potentially indicating a more aggressive but shorter-lived approach to these attacks.

To further analyze differences between brands, we conduct Mann-Whitney U tests (all p-values < 0.001, post-Bonferroni correction). The starkest contrast between USPS and Instagram indicates vastly different phishing lifespan distributions. Facebook and Instagram, despite longer median lifespans (8.68 and 12.62 hours, respectively), display distinct patterns in their persistent sites, suggesting unique endurance characteristics. Further statistics are available in Table 8. **Analysis of Outliers (Longer-lived Phishing).** Our analysis of the longer-lived phishing sites, defined as those lasting beyond each brand's median lifespan, reveals significant brand variations (see Appendix A). Facebook emerges as the most prominent target, with 23% of its attacks falling into this outlier category, followed by USPS at 9.52%. While DHL-themed sites are less common, comprising only 1.08% of longer-lived incidents, they show remarkable persistence, lasting an average of 122.45 hours (5.10 days) and reaching a maximum duration of 750.26 hours (31.26 days) among top brands. Although Instagram is related to 1.48% of longer-lived websites, its persistent cases have a notably high median lifespan of 32.22 hours. AT&T presents an interesting contrast, with 2.90% of sites in this category but the shortest average duration at 78.8 hours among the longer-lasting group.

> **Takeaway 1:** The average lifespan of phishing websites is 54.04 hours (2.25 days), with significant variations across brands (41.15 to 73.85 hours). While most sites are taken down quickly (median 5.46 hours), a concerning subset persists for extended periods. Logistic companies tend to operate within a compressed time frame than other sectors. Our analysis identifies 49 unique brands with at least one phishing site lasting 30 days or more. We find significant differences in phishing lifetime across all brand pairs and variability in blocking effectiveness across brands.

## 5.2 Effectiveness of Google Safe Browsing

We scrutinize phishing lifespan related to GSB [4] through the following analyses (see Table 2): ➊ the number of phishing attempts detected and undetected by GSB, ➋ GSB detection time differences between typical and redirected phishing URLs, ➌ point-in-time cases detected by GSB (based on APWG data and phishing site activity status), and ➍ phishing attempts that exceeded the average GSB detection time.
**GSB Detection Rate.** We find that GSB includes 18.41% (52,696) of the total phishing URLs in our dataset. In other words, 81.59% of the phishing URLs are not blocklisted by GSB, meaning users are exposed to such phishing websites without GSB's intervention. We further analyze how effectively GSB detects phishing websites with the redirection phishing evasion technique, which uses benign URLs that eventually redirect to the final phishing website. Such benign URLs can be easily changed during the phishing campaign, hiding the final phishing websites on the back. Due to the volatility of those redirection URLs, blocklisting them is often ineffective.

We find that 32,745 (11.44%) URLs are typical phishing URLs (*i.e.*, no redirection), while 19,951 (6.97%) are redirected URLs, demonstrating GSB's limited capability against the redirection evasion technique. Notably, GSB's performance on redirected URLs significantly outpaces seven other popular anti-phishing blocklists

**Table 2: Phishing Site Lifespans and GSB Detection Time.**

| Category | Average | | Median | |
|---|---|---|---|---|
| ➊ *Lifespan Analysis* | | | | |
| GSB Detected (18.41%) | 55.89h | (2.33d) | 7.86h | (0.33d) |
| GSB Not Detected (81.59%) | 35.96h | (1.50d) | 2.67h | (0.11d) |
| ➋ *GSB Detection Times* | | | | |
| Typical URLs (11.44%) | 68.27h | (2.84d) | 4.89h | (0.20d) |
| Redirected URLs (6.97%) | 130.89h | (5.45d) | 5.73h | (0.24d) |
| Total URLs (18.41%) | 108.73h | (4.53d) | 5.80h | (0.24d) |
| ➌ *Detection Scenarios* | | | | |
| GSB detection before APWG | 375.11h | (15.63d) | 28.62h | (1.19d) |
| GSB detection after APWG | 166.69h | (6.95d) | 0.81h | (0.03d) |
| APWG & GSB detect active phishing | 223.58h | (9.32d) | 20.51h | (0.85d) |
| GSB detection after phishing is down | 404.19h | (16.84d) | 23.45h | (0.98d) |
| ➍ *Long-tail Analysis (>4.5 days)* | | | | |
| GSB Detected | 274.93h | (11.46d) | 236.85h | (9.87d) |
| GSB Undetected | 273.58h | (11.40d) | 260.69h | (10.86d) |

(malware-filter [32], OpenPhish [33], Phishing Army [34], Phishing Database [35], Phishunt [36], PhishStats [37], and PhishTank [38]), as detailed in Appendix B.

We conduct an in-depth case study to examine GSB's detection capabilities against phishing campaigns with intensive redirection, *i.e.*, phishing attackers using multiple redirection URLs to a single destination domain. We identify 1,994 groups of such cases, *i.e.*, multiple redirection URLs toward the same destination, encompassing a total of 9,581 unique source URLs. GSB detects 189 of these groups, but only 29 of these detections included the redirected (destination) URLs. This suggests that GSB's effectiveness is limited in dealing with complex redirection schemes used by phishing attackers.
**Lifespan Analysis with GSB.** Our ➊ lifespan analysis reveals significant differences between phishing sites detected by GSB and the others not detected. GSB successfully detects 18.41% of the phishing sites in our dataset, which have an average lifespan of 55.89 hours (2.33 days) and a median lifespan of 7.86 hours (0.33 days). In contrast, the majority of phishing sites (81.59%) not detected by GSB have shorter lifespans, averaging 35.96 hours (1.50 days) with a median of 2.67 hours (0.11 days). This suggests that while GSB detects only a small portion of phishing sites, the detected ones tend to have longer lifespans than undetected ones.

We find that phishing websites without the redirection evasion technique have shorter lifespans (mean of 4.5 days, median of 5.8 hours) than those using redirection (mean of 5.5 days, median of 5.7 hours). This may imply that the redirection technique is effective in extending the operational duration of phishing sites. Our statistical analysis also confirms significant differences in lifespans between typical phishing websites and those employing evasion techniques. Specifically, typical URLs have a significantly higher likelihood (8.24 times higher probability) of being detected by GSB compared to redirected URLs ($p < 0.001$, corrected by FET [39]). Such results suggest that the redirection technique is potentially helping phishing campaigns, which GSB (and the phishing protection technique in general) may be worth paying attention to.

GSB-detected phishing sites show brand-specific lifespan variations. DHL-targeted sites persist the longest (61.98 hours), while AT&T-focused sites average 41.29 hours. USPS-targeted sites anomalously have a 1.76-hour median lifespan but a 30.5-hour GSB detection time, exemplifying skewed phishing site duration distribution.

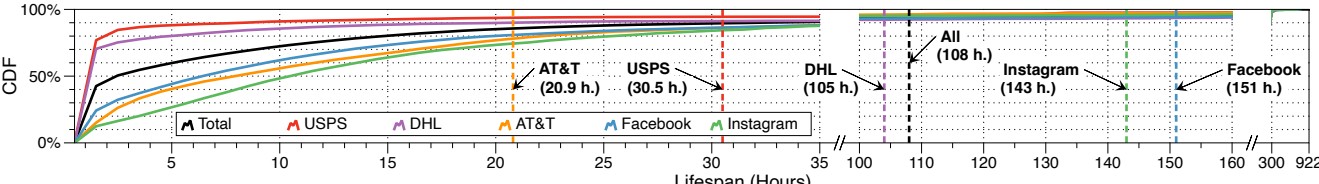

**Figure 2: Lifespan CDF with Top 5 brands. Vertical bars indicate detection time by Google Safe Browsing.**

GSB's 108.73-hour average detection time lags behind 83.93% of phishing sites' takedowns across brands. The remaining 16.07% survive 172.81 hours post-detection on average, ranging from 142.35 hours for AT&T to 202.18 hours for DHL sites.

**GSB Detection Time.** Overall, the mean GSB detection time for all URLs is 108.73 hours (4.53 days), with a median of 5.80 hours. Compared to the average lifespans of phishing websites (54.04 hours), the GSB detection time is slower, often occurring after phishing sites have already been taken down.

We further compare how promptly GSB can detect phishing attacks with evasion techniques (*e.g.*, redirection). We find that GSB may fail to provide timely protection for phishing websites using evasion techniques, as the average detection time for the phishing sites in our dataset represents a considerable exposure period for potential victims. Specifically, our analysis (② GSB detection time) reveals that typical URLs are detected in an average of 68.27 hours (2.84 days), with a median detection time of 4.89 hours, while redirected URLs show a longer average detection time of 130.89 hours (5.45 days) with a median of 5.73 hours.

**GSB Detection Time by Target Brand.** As illustrated in Figure 2, GSB has varying detection times between target brands. Specifically, AT&T-themed phishing sites are detected fastest by GSB, with an average detection time of 20.9 hours. In contrast, Facebook-themed phishing sites take the longest to detect, with an average detection time of 151 hours (about 6.3 days), despite Facebook being the most common target brand in our dataset. Other brands fall between these extremes, with USPS at 30.5 hours, DHL at 105 hours, and Instagram at 143 hours. This means that GSB has different detection times for different phishing target brands.

**Detection Scenarios.** Our analysis of ③ GSB detection scenario in relation to APWG identification and site takedown reveals three distinct scenarios: (1) GSB detects before APWG, (2) GSB detects after APWG, and (3) GSB detects after phishing site takedown. These scenarios show significant variations in phishing site lifespans. When GSB detects before APWG, sites persist for an average of 375.11 hours (15.63 days). In cases where GSB detects after APWG, the average lifespan decreases to 166.69 hours (6.95 days). The average time between APWG and GSB detection is 223.58 hours (9.32 days). Most concerning is when GSB detects after-site deactivation, with an average lifespan of 404.19 hours (16.84 days). This scenario indicates a significant real-time detection gap. Early GSB detection does not always ensure shorter lifespans, possibly due to blocking delays or sophisticated evasion techniques employed by phishers.

**Long-tail Analysis of GSB Detection Time.** The ④ long-tail analysis shows an interesting pattern in the persistence of these threats, focusing on phishing sites that remain active for more than the GSB average detection time of 4.5 days. The 16.44% of phishing sites detected by GSB have an average lifespan of 274.93

hours (11.46 days) and a median of 236.85 hours (9.87 days). In comparison, the 11.01% of phishing sites not detected by GSB have a similar average lifespan of 273.58 hours (11.40 days) but a slightly higher median of 260.69 hours (10.86 days). These results indicate that a significant number of phishing sites persist for long periods, regardless of whether GSB detects them.

> **Takeaway 2:** GSB's phishing detection shows limitations, identifying only 18.41% of sites in 4.5 days on average. 83.93% are blocklisted after takedown, while others survive 7.2 more days. Phishing sites using redirection evasion typically have longer detection times. These findings suggest room for improvement in detection speed and coverage.

## 6 Take-down Causes of Phishing Attacks (RQ2)

To address our second research question **"What factors cause phishing websites to be taken down?"**, we investigate the dataset to identify potential factors that may cause the takedown of phishing websites. In particular, we focus on HTTP response error codes and DNS configurations by analyzing error logs generated during accessing each phishing site. For DNS configuration, we use a combination of the public suffix list [40] and reverse DNS to distinguish between web hosting (*e.g.*, wix.com) and self-hosted websites. From the 90,356 phishing domains across the top ten major brands, we find three primary error types: (1) DNS resolution errors, (2) page not found errors, and (3) timeout errors. 'DNS resolution failure' error occurs when a domain name cannot be resolved to IP addresses. 'Page not found' error arises when requested resources do not exist on servers, while 'timeout' errors result when servers take too long to respond. Each of these presents unique patterns across web hosting and self-hosted environments.

### 6.1 General Causes of Takedowns

Table 3 and Table 9 summarize our results of takedown causes. 'DNS resolution failures' emerge as the predominant issue, accounting for 67.23% of all errors (60,913 domains). The second most common errors are 'page not found' errors, affecting 27.19% of domains (24,573), while 'timeout' errors occurred in 5.39% of cases (4,870).

The distribution of these errors varies significantly across web-hosting and self-hosted environments, as well as among different targeted brands. 'DNS resolution failure' errors vary significantly across phishing attack types. For web-hosting sites, these errors range from 35.42% in Google-themed phishing to 82.95% in Rakuten-themed phishing. Similarly, for self-hosted phishing sites, DNS resolution failures range from 43.37% for DHL-themed phishing to 78.46% for Google-themed phishing. 'Page not found' errors also show notable variability. In web-hosting sites, these errors range from 9.09% in Rakuten-themed phishing to 53.09% in AT&T-themed

**Table 3: Phishing Site (Root Domain) Analysis by Error Type, Hosting Method, and Brands.**

| Error Type | Hosting Method | Top 10 Phishing Brand (% of errors) | | | | | | | | | | All Brands |
|---|---|---|---|---|---|---|---|---|---|---|---|---|
| | | FB (Meta) | USPS | AT&T | WhatsApp | DHL | Instagram | SwissPass | Evri | Google | Rakuten | |
| **DNS Resolution** | Web Hosting | 12,025 (60.05%) | 605 (73.33%) | 1,577 (44.11%) | 796 (64.35%) | 411 (60.35%) | 1,093 (62.71%) | 719 (61.40%) | 68 (69.39%) | 17 (35.42%) | 73 (82.95%) | 26,765 (66.64%) |
| | Self-Hosted | 5,984 (75.39%) | 8,071 (60.94%) | 282 (61.84%) | 1,516 (71.27%) | 1,098 (43.37%) | 205 (64.26%) | 267 (66.75%) | 613 (61.73%) | 397 (78.46%) | 285 (73.26%) | 34,148 (67.81%) |
| **Page Not Found** | Web Hosting | 7,072 (35.32%) | 168 (20.36%) | 1,898 (53.09%) | 274 (22.15%) | 240 (35.35%) | 555 (31.84%) | 395 (33.73%) | 25 (25.51%) | 25 (52.08%) | 8 (9.09%) | 11,442 (28.34%) |
| | Self-Hosted | 2,566 (32.33%) | 4,150 (31.33%) | 155 (33.99%) | 336 (15.79%) | 1,363 (53.85%) | 100 (31.35%) | 98 (24.50%) | 312 (31.42%) | 36 (7.11%) | 80 (20.57%) | 13,131 (26.05%) |
| **Timeout** | Web Hosting | 881 (4.40%) | 49 (5.94%) | 89 (2.49%) | 164 (13.26%) | 28 (4.11%) | 93 (5.33%) | 51 (4.35%) | 5 (5.10%) | 6 (12.50%) | 7 (7.95%) | 2,148 (5.36%) |
| | Self-Hosted | 334 (4.21%) | 991 (7.48%) | 17 (3.73%) | 273 (12.83%) | 64 (2.53%) | 8 (2.51%) | 33 (8.25%) | 67 (6.75%) | 73 (14.43%) | 22 (5.66%) | 2,722 (5.40%) |
| **Total** | | 28,862 (31.85%) | 14,034 (15.49%) | 4,018 (4.43%) | 3,359 (3.71%) | 3,204 (3.54%) | 2,054 (2.27%) | 1,563 (1.73%) | 1,090 (1.20%) | 554 (0.61%) | 475 (0.52%) | 90,356 (99.73%) |

phishing. For self-hosted sites, the range is from 7.11% in Google-themed phishing to 53.85% in DHL-themed phishing.

## 6.2 Taxonomy of Takedown Cause

Our analysis reveals three primary categories of errors leading to phishing site takedowns: DNS resolution failure, page not found error, and timeout errors. Each category shows distinct patterns across web-hosting service and self-hosted environments and among different targeted brands.

**DNS Resolution Failures.** 'DNS resolution failures' dominate the observed takedown causes, accounting for 67.23% of all errors. This prevalence suggests that DNS configuration is notable in phishing site lifecycles. The wide range of DNS resolution failure error rates across brands and hosting types (from 35.42% to 82.95%) indicates significant variability in DNS-related takedowns. For web-hosted sites, Rakuten-themed phishing shows the highest DNS resolution failure error rate (82.95%), while Google-themed phishing has the lowest (35.42%). With websites using self-hosting services, Google-themed phishing exhibits a high rate (78.46%), contrasting with DHL-themed phishing at the lower end (43.37%). These variations suggest that DNS-related takedowns differ substantially based on the targeted brand and hosting method.

**Page Not Found Errors.** 'Page not found' errors are the second most common takedown cause at 27.19%. The similarity in error rates between web-hosting (28.34%) and self-hosting service (26.05%) environments is noteworthy. Among sites using web-hosting services, AT&T-themed phishing has the highest rate (53.09%), while Rakuten-themed phishing shows the lowest (9.09%). For self-hosted sites, DHL-themed phishing tops the list (53.85%), with Google-themed phishing at the bottom (7.11%). These brand-specific variations indicate that the 'page not found' errors are not uniformly distributed across different phishing targets.

**Timeout Errors.** 'Timeout' errors, which are navigation and network timeouts, while less frequent at 5.39%, show consistency between web-hosting (5.36%) and self-hosting service (5.40%) environments. WhatsApp-themed phishing sites stand out with notably higher timeout rates in both web-hosted (13.26%) and self-hosted (12.83%) environments. In contrast, AT&T-themed sites have much lower rates (2.49% web-hosting service, 3.73% self-hosted). These differences indicate that certain brands may be associated with higher rates of timeout errors in phishing attacks.

> **Takeaway 3:** Phishing websites primarily cease operations due to DNS resolution errors (67.23%), suggesting that DNS modification plays a crucial role in phishing lifecycles. The substantial presence of 'page not found' errors (27.19%) indicates a mix of content removal, and a potentially significant portion of takedowns may be phisher-initiated.

## 7 Phishing Volatility in the Lifespan (RQ3)

We answer the third research question, **"What technical measures do phishing websites employ to avoid detection, and how do these change over a website's lifetime?"**, by analyzing screenshots, DNS Records, HTTP headers, and content of collected phishing site resources to track the volatility of phishing websites.

### 7.1 Visual Component Changes (Screenshots)

We analyze how phishing sites' visual representation changes over their lifespan by clustering screenshots based on similarity. Screenshots are collected at each visit, and visual features are extracted to capture the evolution of each site. This process helps us understand how phishing attackers modify the appearance of phishing sites to evade detection. Each screenshot is resized, normalized, and processed to create a feature vector representing its visual content. We detect stable periods and significant visual changes by calculating cosine similarity between screenshots. Screenshots with a similarity score above 0.95 are clustered together. Hence, if more than two clusters are found within each phishing campaign, it may indicate that the phishing website's appearance has changed during the campaign.

**Result.** Figure 3 provides the statistics on the lifespan of phishing sites that experience visual changes during their operation. The data encompasses 71,665 phishing sites, categorized based on the frequency of changes they experience. The analysis reveals that phishing sites with more frequent visual changes tend to have longer lifespans, as reflected in the median and average values. For instance, phishing sites with fewer changes (1 time) have a median lifespan of 1.93 hours and an average lifespan of 9.95 hours. In contrast, sites that experience more changes (50-99 times) exhibit a significantly longer median lifespan of 38.18 hours and an average of 76.49 hours. The longest-living phishing sites, those with over 100 changes, have a median lifespan of 404.52 hours (about 17 days) and an average of 358.35 hours, highlighting how frequent changes help phishing sites evade detection for extended periods.

The quartile statistics (Q1 and Q3) further emphasize this trend. Sites with fewer changes have lower quartile ranges, such as "1 time," with a Q1 of 0.48 hours and a Q3 of 5.80 hours, indicating that most of these sites are short-lived. However, for phishing sites with over 100 changes, the Q1 jumps to 240.30 hours, and the Q3 reaches 480.75 hours, demonstrating that many of these sites persist for

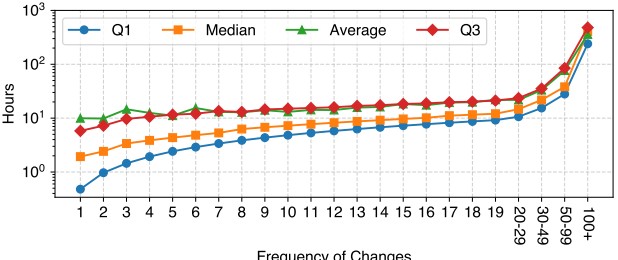

**Figure 3: Phishing Visual Component Frequency of Changes. The X-axis indicates the lifespan, and the Y-axis indicates the number of appearance changes. It reveals that phishing sites with more visual changes tend to have longer lifespans.**

long durations before being detected or taken down. These findings underscore the effectiveness of visual changes in prolonging the operational lifespan of phishing sites (see Appendix F).

---

**Takeaway 4:** Phishing sites frequently changing their visual appearance tend to have significantly longer lifespans. Sites with over 100 visual changes persist for a median of 17 days, compared to 1.93 hours for sites with only one change. This demonstrates that visual modifications are effective for evading detection and prolonging phishing campaigns.

---

## 7.2 DOM Changes from Discovery to Takedown

We analyze changes in website resources between the discovery and takedown of 286,237 phishing sites, with 10.5% (30,069) showing final changes. Our analysis reveals statistically significant changes ($p < 0.001$, t-test) across various metrics, as shown in Table 4. These changes indicate that phishing sites actively evolve their structure and behavior over time, likely in an attempt to evade detection or improve the sites.

The structural complexity of phishing sites is reduced between discovery and takedown. The total number of DOM elements, maximum DOM depth, and third-party scripts decrease by 11.58%, 6.27%, and 12.37%, respectively. This reduction in complexity suggests that phishers may be simplifying their sites, possibly to improve loading speeds or to appear less suspicious to detection systems. In anti-detection techniques, Canvas fingerprinting shows the most significant increase of 37.09%, while browser plugin detection and screen resolution checks decrease by 23.08% and 26.92% respectively. The increase in Canvas fingerprinting usage indicates a shift towards more sophisticated visitor tracking methods, which could potentially enhance phishers' ability to identify victims.

We observe changes in dynamic content and obfuscation usage. The use of AJAX decreases by 27.53%, representing the largest decline in this category. Conversely, the use of event listeners increases by 6.11%. Other notable changes include a decrease in potential Base64 usage (-6.69%) and an increase in potential Unicode escapes (+6.03%). These shifts suggest a move away from server-side dynamic content (AJAX) towards more client-side interactivity (event listeners), which could make the sites harder to analyze using static methods. Obfuscation techniques demonstrated opposite trends for JavaScript and CSS. JS obfuscation decreased by 9.30%, while CSS obfuscation increased by 3.73%. This change in obfuscation strategy might reflect an attempt to evade detection by moving

**Table 4: Comprehensive Changes in Phishing Resources.**

| Metric | Discovery | Takedown | Change |
|---|---|---|---|
| ***Site Complexity*** | | | |
| Avg. Total Elements | 211.06 | 186.62 | -24.44 (-11.58%) |
| Avg. Max Depth | 11.17 | 10.47 | -0.70 (-6.27%) |
| Avg. Third-Party Scripts | 2.91 | 2.55 | -0.36 (-12.37%) |
| ***Anti-detection Techniques (%)*** | | | |
| User Agent Checks | 12.39 | 12.43 | +0.04 (+0.32%) |
| Canvas Fingerprinting | 4.53 | 6.21 | +1.68 (+37.09%) |
| WebGL Fingerprinting | 0.13 | 0.11 | -0.02 (-15.38%) |
| Browser Plugin Detection | 0.13 | 0.10 | -0.03 (-23.08%) |
| Screen Resolution Checks | 1.04 | 0.76 | -0.28 (-26.92%) |
| ***Dynamic Content and Obfuscation (%)*** | | | |
| Uses AJAX | 10.79 | 7.82 | -2.97 (-27.53%) |
| Dynamic DOM Manipulation | 26.42 | 25.43 | -0.99 (-3.75%) |
| Uses Event Listeners | 27.19 | 28.85 | +1.66 (+6.11%) |
| Potential Eval Usage | 4.11 | 4.09 | -0.02 (-0.49%) |
| Potential Base64 Usage | 62.31 | 58.14 | -4.17 (-6.69%) |
| Potential Hex Encoding | 12.67 | 12.38 | -0.29 (-2.29%) |
| Potential Unicode Escapes | 14.09 | 14.94 | +0.85 (+6.03%) |
| ***Obfuscation Techniques (%)*** | | | |
| JS Obfuscation | 5.16 | 4.68 | -0.48 (-9.30%) |
| CSS Obfuscation | 12.59 | 13.06 | +0.47 (+3.73%) |

∗ All changes are statistically significant ($p < 0.001$).
† Percentages in parentheses show relative change.

some obfuscation from JavaScript, which is often closely scrutinized, to CSS, which may receive less attention from detectors.

These quantitative changes emphasize the dynamic nature of phishing websites before they are discovered and blocked. Statistically significant changes in multiple parameters indicate that a phishing site undergoes significant modifications during its lifetime.

## 7.3 DNS Configuration Changes in Phishings

Our analysis of DNS configurations in phishing websites reveals significant variations and changes between the discovery and takedown phases. We examine DNS information associated with phishing websites' IP addresses, performing both forward and reverse DNS lookups to understand how these configurations evolve. DNS usage patterns vary considerably across different brands targeted by phishing attacks. For instance, Facebook-themed phishing predominantly relies on `1e100.net` (35.43% at discovery, increasing to 36.18% at takedown) and `github.com` (29.10% at discovery, decreasing to 27.81% at takedown). In contrast, USPS-themed phishing mainly utilizes `cloudfront.net`, with usage increasing slightly from 48.76% at discovery to 49.46% at takedown. We observe fluidity in DNS services across phishing websites. 7.24% of the sites exhibit changes in their DNS services during their lifetime, with 2.61% adding services, 2.73% removing services, and 1.90% switching services. This fluidity suggests active management of DNS configurations by phishers.

We uncover significant modifications in DNS record configurations, with a clear trend towards shorter time-to-live (TTL) durations. Our analysis reveals that 75.84% of phishing websites alter their DNS settings between discovery and takedown. These values span from 0 to 604,800 seconds, with an average of 1,559.87 seconds and a median of 295 seconds. Notably, the majority of these configurations are set to brief intervals: 81.44% at discovery and 81.89% at takedown are below 1800 seconds. Even more striking, 71.90% fall under 600 seconds, and 51.10% are less than 300 seconds.

This tendency towards shorter durations intensifies over time, with the mean value decreasing from 1,705.63 seconds at discovery to 1,383.54 seconds at takedown. In extreme cases, some are set to 0 seconds, effectively disabling caching. These short-lived DNS configurations can enable fast-flux networks, a technique observed in sophisticated phishing operations [18, 41]. Such changes, particularly using brief and decreasing durations, align with known strategies for evading detection and complicating efforts to track and block malicious infrastructure [42].

## 7.4 Phishing Server Transition

We analyze server information from HTTP headers and track changes in server configurations on phishing websites, focusing on the widely used Apache and Nginx servers [43]. Our findings show that 13 websites using Nginx change versions over time, with 12 upgrading to newer versions and one downgrading. Notably, 2 of the upgraded websites switch to lower minor versions. Similarly, four websites using Apache changed versions, with one downgrading to a lower patch version. Phishing websites often update their server configurations, typically upgrading to newer versions, but occasional downgrades to lower versions suggest potential security weaknesses that could be exploited for detection and mitigation.

> **Takeaway 5:** Our analysis reveals that 30,069 phishing sites undergo final changes before takedown, demonstrating significant volatility in their characteristics over time. Notably, while most features decreased, canvas fingerprinting showed a substantial increase of 37.09%, suggesting an evolution in anti-detection techniques. The contrasting trends in obfuscation methods, with JS obfuscation decreasing by 9.30% while CSS obfuscation increasing by 3.73%, indicate a shift in how phishing sites attempt to conceal their behavior.

## 8 Discussion

**Limitation.** Our study relies on APWG URLs, and there is a time frame during which victims may visit phishing websites before APWG detects the URLs. However, previous research [5] suggests that this time frame is minimal. Thus, while this limitation exists, it likely has a negligible impact on our conclusions regarding phishing activity patterns.

**Recommendations.**

- Given that GSB cannot detect 81.59% of phishing URLs in our dataset, there is a clear need for more robust detection methods. We recommend developing advanced machine learning models incorporating various features, including visual elements, domain characteristics, and content analysis. Continuous monitoring in real-time systems can help detect subtle changes in website behavior over time. Additionally, implementing automated takedown processes would enable rapid removal of detected phishing sites, reducing the exposure for potential victims.

- We find that 16.07% of phishing sites persist for an additional 7.2 days even after being flagged by GSB. Anti-phishing efforts should focus on continuous detection and rapid response mechanisms to address this issue. This approach involves implementing systems that constantly monitor flagged sites for infrastructure, content, or behavior changes. Continuous detection allows us

to identify attempts to evade blocking measures and adapt our defenses accordingly.

**Ethics.** Our data collection process adheres to strict ethical guidelines. We block form submissions on phishing sites using customized Puppeteer event listeners. All data is anonymized with one-way hashing and securely stored behind robust firewalls and access controls. Any inadvertently captured personal data is immediately scrubbed using regularly updated regex patterns and entity recognition models. Automated scripts enforce data retention policies, and access is restricted to authorized researchers only. We disclose our findings to GSB.

## 9 Related Work

Previous research has largely neglected the analysis of the phishing ecosystem, with earlier studies primarily concentrating on understanding phishing attacks with detection mechanisms.

**Phishing Ecosystem.** Previous research [5, 13, 20] has explored phishing attacks through controlled experiments. Moreover, additional studies [8, 44, 45] have investigated current mitigation strategies by analyzing how existing detection mechanisms function. These works focus on understanding the overall structure of phishing websites and their evasion tactics. In contrast, our research aims to delve deeper into the ecosystem with a lifetime of phishing attacks and a longitudinal understanding of the phishing ecosystem evolution.

**Phishing Lifecycle.** Research has examined the period between phishing domain registration and when these domains are eventually detected and blocklisted [6, 7], providing insights into how attackers attempt to prolong the lifespan of their phishing websites. Additionally, honeypots have been used to analyze phishing campaigns, capturing phishing attempts in real-time and revealing attacker tactics and strategies [8]. Another important area of study explores victim traffic on phishing websites, particularly in financial organizations, to understand how users interact with these fraudulent sites and how attackers exploit this traffic before detection [5]. Compared to previous work, our study focuses on a more comprehensive analysis of the phishing lifecycle, examining the various factors that contribute to the duration and evolution of phishing websites.

## 10 Conclusion

Phishing websites remain active for an average of 2.25 days, and their short-lived nature reduces the effectiveness of blocklist-based defenses. GSB takes an average of 4.5 days to detect these sites, indicating many phishing operations terminate before detection. Moreover, our analysis reveals that 16.07% of sites persist for an additional 7.2 days post-detection. Widespread evasion techniques significantly hinder traditional detection methods, including short DNS TTL values and frequent visual changes. Phishing sites with extensive visual changes (100+) exhibited a median lifespan of 17 days, while those with minimal modifications lasted only 1.93 hours. The heavy reliance on popular hosting services further complicates mitigation efforts. These findings highlight the limitations of blocklist-based approaches and emphasize the need for sophisticated detection methods. To combat rapidly evolving phishing attacks, real-time and adaptive defense mechanisms are essential.

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

# A  Comparison of Loger-lived Phishing Brands.

Our analysis of longer-lived phishing sites, defined as those lasting beyond each brand's median lifespan, reveals significant variations across targeted brands (Table 5). While less common, DHL-themed sites show remarkable persistence, with the highest mean duration of 122.45 hours (approximately 5.10 days) and a maximum lifespan of 750.26 hours (about 31.26 days). Facebook-themed sites, being more prevalent, exhibit a high mean lifespan of 109.77 hours (about 4.57 days) and the highest recorded maximum

of 799.52 hours (approximately 33.31 days). Despite lower overall phishing incidence, Instagram shows a notably high median lifespan of 32.22 hours for its persistent cases. USPS-themed sites present an interesting case. They have the lowest median lifespan (6.95 hours) among longer-lived sites but still maintain a substantial mean duration of 86.72 hours, indicating a skewed distribution with persistent outliers. AT&T exhibits the shortest mean lifespan (78.83 hours) among the longer-lasting group.

**Table 5: Comparison of Longer-lived Top 5 Phishing Brands.**

| Metric (hours) | USPS | DHL | Facebook | AT&T | Instagram |
|---|---|---|---|---|---|
| Mean | 86.72 | 122.45 | 109.77 | 78.83 | 102.67 |
| Median | 6.95 | 18.59 | 29.92 | 24.74 | 32.22 |
| 90th Percentile | 319.62 | 351.28 | 304.11 | 250.06 | 295.65 |
| 95th Percentile | 353.36 | 535.42 | 350.61 | 324.42 | 341.59 |
| Max | 747.57 | 750.26 | 799.52 | 751.09 | 744.15 |

## B  Result of Anti-phishing Blocklists

As discussed in Section 5.2, GSB detected 18.41% of the phishing URLs in our dataset. However, our analysis of seven other popular anti-phishing blocklists reveals significantly lower detection rates, as shown in Table 6. These results underscore the challenges of anti-phishing blocklists in keeping pace with the rapidly evolving phishing landscape. The stark contrast between GSB's performance and other blocklists suggests that GSB may employ more sophisticated detection methods or access a broader range of data sources. The generally lower detection rates for redirect URLs across all blocklists indicate that phishers use redirection techniques to evade detection effectively. This aligns with our findings in Section 7 regarding the prevalence of redirection in phishing attacks.

**Table 6: Comparison of 7 Blocklist Phishing URLs.**

| Blocklist | Typical URL | | Redirected URL | |
|---|---|---|---|---|
| | Matches | Percentage | Matches | Percentage |
| malware-filter [32] | 3,600 | 1.26% | 2,400 | 0.84% |
| OpenPhish [33] | 9,525 | 3.33% | 7,163 | 2.50% |
| Phishunt [36] | 761 | 0.27% | 576 | 0.20% |
| Phishing Army [34] | 756 | 0.26% | 424 | 0.15% |
| Phishing Database [35] | 10 | 0.00% | 9 | 0.00% |
| PhishStats [37] | 1 | 0.00% | 0 | 0.00% |
| PhishTank [38] | 0 | 0.00% | 0 | 0.00% |

## C  Usage of ASNs and TLDs in Phishing Sites

Our analysis of the distribution of Autonomous System Numbers (ASNs) and Top-Level Domains (TLDs) in phishing sites reveals interesting patterns in the infrastructure used by attackers. As shown in Table 7, Cloudflare emerges as the dominant ASN, hosting 53.68% of the phishing sites in our dataset. This is followed by a significant number of sites (7.68%) with unknown ASNs and then by major cloud providers such as Google (4.55%), Alibaba (4.18%), and Amazon (3.85%). These findings suggest that phishers often leverage popular cloud and CDN services to host their malicious content, potentially benefiting from these platforms' reliability and ability to obfuscate the true origin of the attacks.

In terms of TLDs, while the traditional '.com' domain remains the most prevalent (28.11%), we observe a notable use of newer or less common TLDs. The '.shop' TLD, for instance, accounts for 24.87% of

the phishing sites, indicating its popularity among attackers. Other frequently used TLDs include '.top' (7.54%), '.dev' (5.83%), and '.io' (3.41%). This diversification in TLD usage may reflect attempts by phishers to evade detection or to create more convincing fake domains that align with their targeted brands or services.

**Table 7: Distribution of ASNs and TLDs.**

| ASN | Name | Number (%) | TLD | Number (%) |
|---|---|---|---|---|
| 13335 | Cloudflare | 153664 (53.68%) | com | 80455 (28.11%) |
| Unknown | Unknown | 21995 (7.68%) | shop | 71200 (24.87%) |
| 15169 | Google | 13025 (4.55%) | top | 21568 (7.54%) |
| 45102 | Alibaba | 11961 (4.18%) | dev | 16684 (5.83%) |
| 16509 | Amazon | 11009 (3.85%) | io | 9754 (3.41%) |
| 54113 | Fastly | 10520 (3.68%) | org | 7157 (2.50%) |
| 132203 | Tencent | 8008 (2.80%) | app | 7104 (2.48%) |
| 133199 | SonderCloud | 7618 (2.66%) | cfd | 6415 (2.24%) |
| 27323 | ServerStadium | 4109 (1.44%) | cn | 5261 (1.84%) |
| 14061 | DigitalOcean | 3626 (1.27%) | me | 4765 (1.67%) |

## D  Statistical Results for Longer-lived Phishing

**Table 8: Mann-Whitney U Test for Longer-lived Phishing.**

| Comparison | U Statistic | $p$-value | Sample Sizes | Significant |
|---|---|---|---|---|
| USPS vs DHL | 8,889,555.5 | 2.20e-78 | 14733, 1674 | Yes |
| USPS vs Facebook | 148,623,111.5 | 0.00e+00 | 14733, 37454 | Yes |
| USPS vs AT&T | 20,392,662.0 | 2.51e-271 | 14733, 4278 | Yes |
| USPS vs Instagram | 9,014,440.0 | 1.40e-274 | 14733, 2270 | Yes |
| DHL vs Facebook | 22,780,213.0 | 4.33e-80 | 1674, 37454 | Yes |
| DHL vs AT&T | 3,052,938.5 | 8.43e-19 | 1674, 4278 | Yes |
| DHL vs Instagram | 1,344,337.5 | 1.09e-55 | 1674, 2270 | Yes |
| Facebook vs AT&T | 92,660,283.5 | 2.17e-63 | 37454, 4278 | Yes |
| Facebook vs Instagram | 38,870,796.0 | 6.88e-12 | 37454, 2270 | Yes |
| AT&T vs Instagram | 3,487,045.5 | 7.86e-79 | 4278, 2270 | Yes |

∗ All $p$-values are Bonferroni corrected; Significance level: $\alpha = 0.05$.

Mann-Whitney U tests [46] compare the lifespan distributions of longer-lived phishing sites across different brands, as shown in Table 8. The results reveal statistically significant differences ($p < 0.05$) between all brand pairs, even after applying the conservative Bonferroni correction for multiple comparisons. This indicates that the persistence patterns for phishing attacks vary substantially depending on the targeted brand.

The most pronounced differences are observed between USPS and Instagram (U = 9,014,440, $p$ = 1.40e-274) and USPS and AT&T (U = 20,392,662, $p$ = 2.51e-271), suggesting markedly different attack persistence strategies for these brands. Interestingly, while Facebook has the largest sample size of a longer-lived span, its comparison with Instagram still shows a highly significant difference ($p$ = 6.88e-12), with the highest p-value among all comparisons. This suggests that while there are statistically significant differences in phishing campaign characteristics between these social media platforms, they may be more similar than to other brands in the study. However, it's important to note that all comparisons showed extremely low p-values ($p < 0.05$), indicating significant differences across all brand pairs in how long-lived phishing attacks persist.

## E  Error Causes

Our analysis of error causes in accessing phishing sites reveals significant patterns across various brands, as shown in Table 9. The most prevalent error by far is DNS resolution failure, accounting for

**Table 9: Summary of Error Causes and Their Frequency based on FQDN.**

| Brand | DNS resolution fail | Page not found | Timeout | Access forbidden | Protocol error | DNS refused | Total |
|---|---|---|---|---|---|---|---|
| All brands | 182,239 (63.67%) | 79,626 (27.82%) | 19,468 (6.80%) | 3,767 (1.32%) | 1,070 (0.37%) | 47 (0.02%) | 286,217 (100.00%) |
| Facebook | 64,933 (83.76%) | 9,375 (12.09%) | 2,302 (2.97%) | 809 (1.04%) | 103 (0.13%) | 1 (0.00%) | 77,523 (100.00%) |
| USPS | 19,374 (60.38%) | 10,142 (31.61%) | 2,089 (6.51%) | 413 (1.29%) | 54 (0.17%) | 17 (0.05%) | 32,089 (100.00%) |
| AT&T | 6,464 (65.88%) | 2,870 (29.25%) | 333 (3.39%) | 115 (1.17%) | 28 (0.29%) | 1 (0.01%) | 9,811 (100.00%) |
| WhatsApp | 5,265 (72.52%) | 951 (13.10%) | 934 (12.86%) | 97 (1.34%) | 13 (0.18%) | 0 (0.00%) | 7,260 (100.00%) |
| Instagram | 3,360 (70.81%) | 1,094 (23.06%) | 245 (5.16%) | 34 (0.72%) | 12 (0.25%) | 0 (0.00%) | 4,745 (100.00%) |

63.67% of all errors across brands. This suggests that most phishing sites become inaccessible due to DNS-related issues, potentially indicating rapid takedown efforts or attackers' use of disposable domains. The second most common error is "Page not found" (27.82%), which could be attributed to content removal or site restructuring by the attackers.

Interestingly, the distribution of errors varies across different targeted brands. For instance, Facebook-related phishing sites show a notably higher rate of DNS resolution failures (83.76%) than the overall average. In contrast, USPS-targeted sites have a higher incidence of "Page not found" errors (31.61%). Timeout errors, while less frequent overall (6.80%), are particularly prominent in WhatsApp-related phishing attempts (12.86%). These variations might reflect differences in anti-phishing strategies employed by various brands or unique characteristics of the phishing campaigns targeting them. The relatively low occurrence of access forbidden errors (1.32%) and protocol errors (0.37%) across all brands suggests that when phishing sites are online, they generally remain accessible.

## F  Visual Component Changes

**Table 10: Changed Phishing Lifespan Statistics (in hours).**

| Freq.* | URLs (%) | Q1 | Med. | Avg. | Q3 | Max | Std. |
|---|---|---|---|---|---|---|---|
| 1 | 6,014 (8.39%) | 0.48 | 1.93 | 9.95 | 5.80 | 720 | 36.08 |
| 2 | 4,356 (6.08%) | 0.97 | 2.42 | 9.84 | 7.25 | 624 | 32.54 |
| 3 | 3,817 (5.33%) | 1.45 | 3.38 | 14.62 | 9.67 | 720 | 39.03 |
| 4 | 3,223 (4.50%) | 1.93 | 3.87 | 12.53 | 10.63 | 576 | 34.86 |
| 5 | 2,918 (4.07%) | 2.42 | 4.35 | 11.12 | 11.60 | 456 | 31.49 |
| 6 | 2,842 (3.97%) | 2.90 | 4.83 | 15.43 | 12.08 | 672 | 43.47 |
| 7 | 4,756 (6.64%) | 3.38 | 5.32 | 13.06 | 13.53 | 504 | 35.04 |
| 8 | 2,367 (3.30%) | 3.87 | 6.28 | 12.83 | 13.05 | 384 | 31.46 |
| 9 | 2,162 (3.02%) | 4.35 | 6.77 | 14.03 | 14.50 | 432 | 34.28 |
| 10 | 2,028 (2.83%) | 4.83 | 7.25 | 13.07 | 14.98 | 333 | 26.89 |
| 11 | 1,995 (2.78%) | 5.32 | 7.73 | 14.22 | 15.43 | 381 | 28.84 |
| 12 | 1,859 (2.59%) | 5.80 | 8.22 | 14.15 | 15.92 | 360 | 27.45 |
| 13 | 1,762 (2.46%) | 6.28 | 8.70 | 15.72 | 16.88 | 408 | 29.13 |
| 14 | 1,733 (2.42%) | 6.77 | 9.18 | 16.11 | 17.37 | 384 | 29.30 |
| 15 | 1,713 (2.39%) | 7.25 | 9.67 | 18.31 | 18.33 | 455 | 34.56 |
| 16 | 2,540 (3.54%) | 7.73 | 10.15 | 17.41 | 18.80 | 408 | 30.13 |
| 17 | 1,545 (2.16%) | 8.22 | 11.12 | 19.44 | 19.75 | 480 | 37.78 |
| 18 | 1,352 (1.89%) | 8.70 | 11.60 | 19.79 | 20.23 | 456 | 36.26 |
| 19 | 1,256 (1.75%) | 9.18 | 12.08 | 21.90 | 21.18 | 504 | 42.08 |
| 20-29 | 9,640 (13.45%) | 10.63 | 14.65 | 21.76 | 23.57 | 624 | 33.60 |
| 30-49 | 7,637 (10.66%) | 15.43 | 21.77 | 33.15 | 35.35 | 720 | 45.55 |
| 50-99 | 4,652 (6.49%) | 28.03 | 38.18 | 76.49 | 84.82 | 893 | 91.61 |
| 100+ | 2,498 (3.49%) | 240.30 | 404.52 | 358.35 | 480.75 | 1440 | 199.69 |
| **Total** | 71,665 (100%) | 3.38 | 14.65 | 30.62 | 28.03 | 1440 | 78.43 |

∗ **Freq.** is stands for the frequency that phishing sites have changed their website layout or target brands.

The data in Table 10 provides insights into the lifespan characteristics of phishing websites that undergo visual component changes. These changes may include alterations to the website layout or shifts in the targeted brands. The analysis covers 71,665 URLs, categorized by the frequency of visual changes they underwent during their lifespan.

A notable trend emerges as the frequency of changes increases. Websites with more frequent changes tend to have longer average lifespans. For instance, URLs with only one change have an average lifespan of 9.95 hours, while those with 15 changes survive an average of 18.31 hours. This trend continues, with URLs experiencing 50-99 changes lasting an average of 76.49 hours and those with 100+ changes persisting for an impressive 358.35 hours on average. This positive correlation between change frequency and lifespan suggests that frequent visual updates might be a strategy employed by phishers to evade detection and prolong their operations.

However, it's important to note the significant variability in lifespans across all categories, as indicated by the large standard deviations. For example, URLs with 100+ changes show a standard deviation of 199.69 hours, highlighting the wide range of outcomes even within this most persistent group. Interestingly, while only 3.49% of URLs fall into this 100+ changes category, they demonstrate remarkably extended lifespans, with a median of 404.52 hours (about 16.9 days). This suggests that a small proportion of highly adaptive phishing sites contribute disproportionately to the overall threat landscape by remaining active for extended periods.

## G  CDN Usage in Phishing Websites

Understanding CDN usage in phishing infrastructures is crucial because these networks obscure the true origin of traffic, making it significantly harder to trace and block malicious activity [47, 48]. For phishing websites, this widespread use of CDNs creates a layer of obfuscation, making it difficult for traditional IP-based blocking or detection mechanisms to identify and mitigate these threats effectively [49].

Our analysis reveals significant insights into using Content Delivery Networks (CDNs) in phishing infrastructures targeting major brands. As shown in Table 11, we observed a 4.13% decrease in total IP addresses used for phishing, indicating a slight contraction in the infrastructure footprint. CDN services are heavily utilized across all examined phishing Websites, with 99.80% of IP addresses associated with some form of CDN. This near-universal adoption of CDNs by phishers presents significant challenges for traditional IP-based blocking strategies. The breakdown of CDN types shows a clear preference for Web Application Firewalls (WAF) at 53.45%, followed by traditional CDN services at 33.57%, and cloud services at 12.78%. Cloudflare dominates the WAF category among providers

**Table 11: CDN Changes and Top Providers based on IP.**

| Type | Provider | Typical | Redirected | Change (%) | |
|------|----------|---------|------------|-----------|------|
| CDN | Google | 15,039,132 | 13,745,800 | -1,293,332 | (-8.60%) |
| | Cloudfront | 128,002 | 130,848 | 2,846 | (+2.22%) |
| | Fastly | 42,311 | 48,535 | 6,224 | (+14.71%) |
| Cloud | AWS | 5,498,797 | 5,299,394 | -199,403 | (-3.63%) |
| | Office365 | 276 | 194 | -82 | (-29.71%) |
| | Oracle | 39 | 25 | -14 | (-35.90%) |
| WAF | Cloudflare | 22,473,347 | 22,171,548 | -301,799 | (-1.34%) |
| | Incapsula | 24 | 25 | 1 | (4.17%) |
| **Total** | | 43,181,928 | 41,396,369 | -1,785,559 | (-4.13%) |

with 22,171,548 IP addresses, despite a 1.34% decrease. Google leads in traditional CDN services with 13,745,800 IPs, showing an 8.60% decrease. AWS is the primary cloud provider with 5,299,394 IPs, experiencing a 3.63% reduction.

**Table 12: CDN Usage Analysis for Top Phishing Brands.**

| Brand | Total IPs | | Change | CDN Type Usage | | |
|-------|-----------|------------|--------|------|-------|------|
| | Typical | Redirected | | CDN | Cloud | WAF |
| Facebook | 13,823,887 | 12,070,911 | -12.68% | 35.87% | 13.93% | 49.74% |
| USPS | 6,317,945 | 6,357,207 | +0.62% | 2.52% | 0.27% | 96.99% |
| AT&T | 7,849,403 | 7,798,646 | -0.65% | 94.99% | 0.56% | 4.36% |
| WhatsApp | 616,768 | 770,939 | +25.00% | 11.90% | 4.20% | 83.44% |
| Instagram | 977,337 | 851,690 | -12.86% | 36.95% | 14.84% | 47.49% |
| DHL | 342,154 | 348,068 | +1.73% | 9.79% | 3.48% | 86.14% |
| SwissPass | 153,368 | 185,328 | +20.84% | 42.95% | 4.11% | 52.38% |
| Microsoft | 131,772 | 119,623 | -9.22% | 10.51% | 5.37% | 83.97% |
| Rakuten | 66,022 | 82,142 | +24.41% | 3.33% | 0.76% | 95.63% |
| **All Feeds** | 43,263,524 | 41,477,792 | -4.13% | 33.57% | 12.78% | 53.45% |

As shown in Table 12, Examining brand-specific patterns reveals interesting trends. Social media and messaging platforms (*e.g.*, Facebook, Instagram, and WhatsApp) show diverse CDN usage patterns. Facebook and Instagram rely heavily on traditional CDN and WAF services, while WhatsApp shows a strong preference for WAF (83.44%) and experienced a 25% increase in total IPs. Delivery services such as USPS and DHL show an overwhelming preference for WAF services, with USPS having the highest WAF usage at 96.99%. Financial and e-commerce platforms exhibit varied patterns. Meta shows a unique pattern with high cloud usage (40.33%) alongside WAF (55.80%), while Rakuten heavily favors WAF (95.63%) and saw a 24.41% increase in total IPs. Technology companies also display distinct preferences, with Microsoft primarily using WAF (83.97%) and AT&T standing out with 94.99% usage of traditional CDN services, deviating significantly from the WAF-centric trend.

These patterns suggest that phishers adapt their infrastructure choices based on the targeted brand, potentially to mimic legitimate traffic patterns or exploit specific vulnerabilities in brand-associated services. The pervasive use of CDNs, especially WAF services, in phishing infrastructure poses significant challenges for detection and mitigation.

