# OpenReview forum: "7 Days Later: Analyzing Phishing-Site Lifespan After Detected"
_ACM.org/TheWebConf/2025/Conference — WWW 2025 Oral_

### Official Review · Reviewer_tW1L · 2024-11-03

**Novelty:** 3
**Technical Quality:** 4

**Review:**

The paper presents an analysis of the lifetime of the phishing websites.
Staring woith a list of phishing URLs (from APWG), the paper explores several questions including:
-how long do they live?
-what factors contribute to the duration of their lives?
-how do they avoid detection?

The paper presents lots of results. In several cases, however, the results are more-or-less relatively expected.
For example, it is more-or-less well known that phishing sites have a short lifespan. Knowing the exact lifespan (much like reported in the paper) is nice, but it hardly adds any significant information.

In some other cases the paper presents interesting results, but lacks some explanation. For example, the paper reports that Google Safe Browsing detects only 18.41% of the phishing web sites listed in APWG. It is not clear why Google has such a low detection rate. It seems trivial for Google to gain access to the list of APWG and thus increase their hit rate to 100%. Given that Google can get access to this QPWG list, it is not clear why they have such a low phishing site detection rate.

As another example, the paper reads: "a potentially significant portion of takedowns may be phisher initiated". Although this is very interesting, the paper does not provide any explanation why phishers took down their own websites.

**Questions:**

The paper reads: a potentially significant portion of takedowns may be phisher initiated. Why phisers took down  their own web sites?

**Reviewer Confidence:**

3: The reviewer is confident but not certain that the evaluation is correct

**Scope:**

3: The work is somewhat relevant to the Web and to the track, and is of narrow interest to a sub-community

---

### Official Review · Reviewer_M5vD · 2024-11-04

**Novelty:** 4
**Technical Quality:** 4

**Review:**

The authors conducted a comprehensive analysis of the lifespan and evolution of phishing websites, emphasizing their survival strategies and evasion techniques. The findings are quite intriguing. They validated the effectiveness of GSB, examined the average lifespan of phishing sites, and identified the factors leading to their takedown. This research addresses a significant issue relevant to both the field and the broader research community, providing valuable insights that could inform future studies. However, there are few concerns that should be addressed. Specifically, the proposed approach heavily relies on data collection and analysis. There are publicly available datasets that could enhance this analysis. Additionally, exploring deep learning and machine learning models alongside GSB could further enrich the research. Overall, the presentation is good enough.

**Questions:**

1. In addition to GSB, there are several recent machine learning and deep learning approaches proposed for addressing the phishing website issue. What are the authors' perspectives on these methods? How might they be beneficial if applied in conjunction with GSB?
2. Why weren't other publicly available datasets utilized in this analysis?
3. What is the impact of longer-lived phishing sites?
4. How will the takedown process work if the phishing site changes dynamically?

**Reviewer Confidence:**

3: The reviewer is confident but not certain that the evaluation is correct

**Scope:**

4: The work is relevant to the Web and to the track, and is of broad interest to the community

---

### Official Review · Reviewer_QzPE · 2024-12-02

**Novelty:** 4
**Technical Quality:** 4

**Review:**

The study offers an in-depth examination of the lifespan and evolution of phishing websites, highlighting their strategies for survival and techniques for evading detection. The findings are both insightful and significant. The researchers evaluated the effectiveness of GSB, analyzed the average lifespan of phishing sites, and identified key factors contributing to their takedown. This work addresses a critical problem of relevance to the field and the broader research community, offering valuable insights that could guide future investigations. However, certain aspects could be improved. The reliance on data collection and analysis in the current approach could be enhanced by incorporating publicly available datasets. Furthermore, integrating machine learning and deep learning models alongside GSB could add further depth to the analysis. Overall, the research is well-presented and meaningful.

**Questions:**

1. How does the method perform against sophisticated phishing tactics like AI-generated content or deepfake phishing sites?
2. How scalable is the proposed high-frequency data collection framework in large- scale, real-world applications?
3. Does the reliance on APWG&#39;s dataset introduce any bias that might affect the representativeness of phishing site behaviors and lifespans?
4. How does the takedown process function when a phishing site undergoes dynamic changes?

**Reviewer Confidence:**

3: The reviewer is confident but not certain that the evaluation is correct

**Scope:**

4: The work is relevant to the Web and to the track, and is of broad interest to the community

---

### Official Review · Reviewer_3fU1 · 2024-12-02

**Novelty:** 4
**Technical Quality:** 4

**Review:**

**Pros**
- Provides interesting insights into several phishing aspects
- Will open-source data collection framework

**Cons**
- Methodology might not accurately capture real phishing URL lifespan
- Further methodology details should be provided (e.g., screenshot handling, detection of specific DOM changes)

Thank you for submitting your work to the Web Conference! Overall, I enjoyed reading this paper and found that it is generally well written and provides some interesting insights on different phishing aspects. However, I have some concerns and questions which I outline in the following.

Initially, I think that the term "lifespan" is rather misleading. If I understand correctly, the authors measure the period of time a phishing URL is responsive *after* having being detected by APWG, which does not necessarily reflect the actual lifespan of the URL. Therefore, the accuracy of the reported findings is strongly dependent on how fast APWG detects new phishing URLs. If that is the case and I have not missed something, this should be explicitly clarified and some parts throughout the paper should be rephrased to reflect this (e.g., RQ1).

In section 4, it is mentioned that a given URL is allowed to have three consecutive failures (DNS, 404s, timeouts etc.) and is then excluded from the dataset. Why is that? I think it would be quite insightful (and maybe even commonplace) if phishing sites were again responsive after some time, especially given that their operators seem to tweak them quite often based on your findings in section 7.

Regarding the GSB evaluation in section 5.2, how many URLs were detected by GSB before APWG? Also, how is the lifespan calculated for these websites?

I also think several more details should be provided for the different volatility aspects analyzed in section 7. For instance, in 7.1, how are screenshots precisely preprocessed to create their feature vectors? How was the 0.95 cosine similarity threshold decided and could it end up *not* clustering similar pages with only slight differences? Also, was the cosine similarity calculated for consecutive screenshots? It would also be interesting to report on example cases where a URL exhibited significant changes and what changes were they, e.g., if they were targeting an entirely different brand. On a more general note, the authors mention that "visual modifications are effective for evading detection and prolonging phishing campaigns". I am not entirely sure whether that is the case. The frequent visual changes in a phishing URL, much like the DOM changes, might simply be fine-tuning by the attacker to improve their page's quality. As such, I think this (and the entire RQ3) should be rephrased similarly to 7.2 ("likely in an attempt to evade detection or improve the sites"). Moreover, for the DOM changes in 7.2, it is not mentioned how were the different techniques identified and measured, i.e., canvas fingerprinting, JS and CSS obfuscation.

Finally, some minor remarks:
- In Figure 3, I think the X-axis and Y-axis descriptions should be swapped in the caption (X-axis: # of changes, Y-axis: lifespan).
- 6.2 is essentially a slightly expanded and repetitive version of 6.1. Maybe these could be merged and create some space for additional experiments from the appendix.

**Questions:**

- How exactly is each URL's lifespan calculated?
- How many URLs were detected by GSB before APWG?
- How were the different techniques pertinent to DOM changes (canvas fingerprinting and obfuscation) detected and measured?

**Reviewer Confidence:**

3: The reviewer is confident but not certain that the evaluation is correct

**Scope:**

4: The work is relevant to the Web and to the track, and is of broad interest to the community